# Identification of Novel Inhibitors of Starch Excess 4 (SEX4)

**DOI:** 10.3390/life14121686

**Published:** 2024-12-20

**Authors:** Damhee Lee, Dongsun Lee, Kyujeong Won, Suhyun Kim, Youngjun Kim

**Affiliations:** Department of Medicinal Bioscience, Nanotechnology Research Center, Konkuk University, Chungju 27478, Republic of Korea

**Keywords:** protein inhibitor, protein phosphatase, starch accumulation, starch decomposition, Starch Excess 4

## Abstract

This study identified several inhibitors of Starch Excess 4 (SEX4), an enzyme in plants’ starch decomposition. Our research aims to inhibit starch breakdown by SEX4 with its potential to significantly impact food security, leading to starch accumulation in plants such as potatoes, sweet potatoes, and significant crops like grains and rice. We recognized potential candidates by screening approximately 1840 chemical compounds using the phosphatase assay against *p*NPP. The IC_50_ values of the selected candidates were determined through the *p*NPP assay and the amylopectin assay, while K_i_ values were confirmed by calculating V_max_, K_M_, and *k*_cat_ values. Finally, we compared the IC_50_ values of Like Sex Four 2 (LSF2) and SEX4 to assess their selectivity. This screening yielded several potential inhibitory compounds, with F05 showing promise in the *p*NPP assay and F09 and G11 in the amylopectin assay, all demonstrating more selectivity for SEX4 than LSF2. Consequently, we identified seven chemicals as promising inhibitor compounds, offering potential for future research and applications. However, further quantitative structure–activity relationship studies and the practical application to test selected compounds on crops will be necessary in future research.

## 1. Introduction

Protein Tyrosine Phosphatases (PTPs) belong to the protein phosphatase family [1,2]. Members of the PTP family usually catalyze the dephosphorylation of protein substrates. However, some phosphatases from a subclass of PTPs, the Dual-Specificity Phosphatases (DSPs), dephosphorylate non-proteinaceous substrates such as lipids and polysaccharides [3,4]. In addition, the glucan phosphatase also belongs to the DSP family, and Starch Excess 4 (SEX4), Like Sex Four 2 (LSF2), and Laforin (the human homolog of SEX4) are included in the glucan phosphatase member [5,6,7,8,9,10,11,12].

Starch is the primary storage carbohydrate reserve in plants [13]. During the day, starch is synthesized as the product of photosynthetic carbon assimilation in Arabidopsis (*Arabidopsis thaliana*) leaves. Starch accumulates in the chloroplasts [14,15]. During the subsequent night, the starch is degraded for growth. The supply of carbohydrates from nighttime starch degradation is essential for plant growth [13,14,16,17]. Starch comprises 70–80% *w*/*w* amylopectin and 20–30% *w*/*w* amylose in Arabidopsis [18]. Amylopectin, similar to glycogen, comprises α-1,4-glycosidic linkages with α-1,6-glycosidic branches. Amylose is a linear molecule composed of glucose moieties linked by α-1,4-glycosidic linkages [19,20].

The primary mechanism of starch degradation is reversible glucan phosphorylation at the starch granule surface in the chloroplasts of plants by dikinases and phosphatases [21]. This mechanism is initiated by α-glucan water dikinase (GWD) and phosphoglucan water dikinase (PWD). GWD phosphorylates the C6 position of starch glycosyl residues, providing pre-phosphorylation priming for PWD to phosphorylate the C3 position [22,23,24].

The phosphorylation of starch by GWD and PWD interrupts the packing of amylopectin double helices, allowing the release of maltose and oligosaccharides by β-amylases (BAM1 and BAM3) and isoamylases (ISA1, ISA2, and ISA3) [22,25,26,27,28,29,30,31,32,33]. LSF2 and SEX4 dephosphorylate C3 and C6 mono-phosphate, respectively, enabling efficient starch breakdown and preventing the accumulation of phosphorylated oligosaccharides [9,34]. Together, these enzymes work in a coordinated manner at the starch granule surface to degrade starch throughout the night [13].

The dephosphorylation of starch by SEX4 is essential for proper starch degradation. Without SEX4 activity, β-amylase activity is inhibited and phospho-oligosaccharides are released [35,36]. Following the loss of SEX4 function, Arabidopsis contains three times more starch than typical Arabidopsis leaves [17]. Moreover, SEX4 mutants accumulate phospho-oligosaccharides, suggesting that these plants have a decreased pool of hydrolysable starch, resulting in stunted growth [37,38].

There are many genes that are involved in starch biosynthesis and degradation in storage organs in plants [39]. The starch synthesis and degradation in plants depends on the diurnal cycle, the post-translational regulation of enzyme activity, and starch phosphorylation [40]. The expression of an engineered Laforin in potato resulted in the significantly higher phosphate content of starch [41]. The repression of SEX4 and LSF2 in potato also increased phosphate-bound starch in tubers [36]. Silencing a gene of starch dephosphorylation in Cassava (*Manihot esculenta* Crantz) can specifically alter certain cassava storage root starch properties, potentially increasing its value in food and non-food industries [42]. Rice SEX4-knockdown plants showed starch accumulation in suspension-cultured cells, leaves, and rice straw with normal plant growth and no yield penalty [43]. Thus, the glucan’s coordinated phosphorylation and dephosphorylation are necessary for proper starch degradation [38]. Because of the starch degradation for growth at night, starch will continue to accumulate in storage organs in plants if there is a properly organized inhibition of starch dephosphorylation.

The protein sequences of SEX4 and LSF2 are conserved in plants, such as Arabidopsis, barley, rice, potato, and Cassava [30,36,42,43]. If the activities of SEX4 and LSF2 in starch degradation are effectively inhibited in Arabidopsis, potato, Cassava, and rice, starch accumulation could occur in Arabidopsis leaves, potato tubers, Cassava root, and rice straw. Therefore, chemical compounds inhibiting the phosphatase activity of SEX4 or LSF2 can be helpful tools for accumulating starch in plants.

These inhibitors could be applied in plants, such as potatoes, sweet potatoes, and significant crops like grains and rice, by searching for effective inhibitor compounds of SEX4 or LSF2, and starch can be accumulated. This could have significant implications for the agricultural industry, potentially leading to increased yields, improved food security, and increased bioethanol production. Therefore, by considering the biological importance of SEX4, we have explored the investigation of selective inhibitors for SEX4 using a biochemical approach.

## 2. Materials and Methods

### 2.1. Materials

The *Arabidopsis thaliana* SEX4 lacking the first 89 amino acids (Δ89-SEX4) and LSF2 proteins [9,44] were kindly provided by Dr. Gentry’s group at the University of Kentucky College of Medicine (Lexington, KY, USA). *para*-nitrophenyl phosphate disodium salt hexahydrate (*p*NPP), Malachite green oxalate salt, ammonium molybdate tetrahydrate, and amylopectin from potato starch were purchased from Sigma-Aldrich. Absorbance was measured on a Beckman Coulter (Brea, CA, USA) DU 700 Series UV/Vis Spectrophotometer. The chemical compound library (~1840 compounds), comprising drug-like and natural-like compounds, was obtained from the Korea Chemical Bank (chembank.org, Daejeon, Republic of Korea).

### 2.2. Phosphatase Assay Against pNPP

The phosphatase activity of SEX4 or LSF2 against *p*NPP in the presence or absence of an inhibitor was measured in this assay. The reaction (20 μL of total volume) of *A. thaliana* SEX4 or LSF2 was performed in the reaction mixture containing 1~2 μg SEX4 or LSF2, 1X assay buffer pH 5.5 (20 mM sodium acetate, 10 mM bis-Tris, 10 mM Tris, pH with HCl), 5 mM Dithiothreitol (DTT), and 50 mM *p*NPP (Sigma-Aldrich, Wien, Austria). It was incubated at 37 °C for 10 min with or without an inhibitor. The reaction was terminated by adding 80 μL of 0.25 N NaOH, and absorbance was measured at 410 nm.

### 2.3. Phosphatase Assay Against Amylopectin

The phosphatase activity of SEX4 or LSF2 toward amylopectin in the absence or presence of an inhibitor was measured in a reaction mixture with 2 μg SEX4 or LSF2 at 37 °C in 20 μL volume. The reaction mixture contained 100 mM sodium acetate, 50 mM bis-Tris, 50 mM Tris (pH 6.5), 10 mM DTT, and 45 μg amylopectin with or without an inhibitor. After 15 min of reaction, the reactions were quenched by adding 20 μL of 100 mM N-ethylmalemide and 40 μL of Malachite green reagent (1.25% *w*/*v* ammonium molybdate tetrahydrate (Sigma-Aldrich, Darmstadt, Germany) and 0.15% *w*/*v* Malachite green oxalate salt (Sigma-Aldrich, Germany) in 1 M HCl) for 40 min. The release of free inorganic phosphate was determined by measuring the absorbance at 620 nm and the K_2_HPO_4_ standard curve.

### 2.4. Determination of Steady-State Kinetic and Inhibition Constants

The steady-state kinetic parameters were determined by varying substrate concentrations (*p*NPP 0.5–50 mM; amylopectin 5–45 μg) with or without an inhibitor. The data were fitted by nonlinear regression to the Michaelis–Menten equation using GraphPad PRISM 9.0 software to derive K_M_ and *k*_cat_ values. IC_50_ was determined by a concentration–response assay, using a fixed substrate concentration and various inhibitor concentrations (0–0.2 mM). Inhibition constants (K_i_) and inhibition types were determined by varying substrate concentrations in the absence of an inhibitor and in the presence of an inhibitor at the IC_50_ concentration.

### 2.5. Identification of an Effective Inhibitor for SEX4

The process of finding an inhibitor proceeded as follows (Figure 1). First, the chemical compound library (~1840 compounds, with a concentration of 0.2 mM), comprising drug-like and natural-like compounds, was screened through a *p*NPP assay. Compounds showing an inhibition rate of more than 50% through the inhibition measurement were initially selected. Then, reproduction experiments were performed with these compounds to select candidate substances. Seven selected and confirmed candidates were subjected to a *p*NPP assay and an amylopectin assay to obtain the IC_50_ values, and the K_i_ values were calculated using the V_max_, K_M_, and *k*_cat_ values through steady-state kinetics to find an effective inhibitor. In order to assess any difference in selectivity between SEX4 and LSF2 for the selected candidates, the IC_50_ values of LSF2 were obtained through the *p*NPP and amylopectin assays and compared with those of SEX4.

## 3. Results

### 3.1. Screening of Compound Library

The chemical compound library (~1840 compounds), comprising drug-like and natural-like compounds, was screened to investigate a potential inhibitor for SEX4. Biochemical screening was performed using 1~2 μg of receptor concentration (SEX4) and 50 mM of substrate (*p*NPP) concentration in the presence (200 μM) and absence of inhibitors, and experimental results were collected from absorbance measurements at 410 nm and are presented in Appendix A.

There were several compounds showing an inhibition rate of more than 50%. We rechecked their activity using the *p*NPP assay after repeated experiments to confirm reproducibility, and selected seven chemicals, which were showing high reproducibility. Except for the seven chemicals, the others were excluded from a candidate list due to solubility problems or low reproducibility.

We eventually obtained a candidate list of seven compounds (F09, H08, F05, G07, G11, D04, and H03) and then named them based on their respective plate positions. The chemical names and structural information of the seven candidate compounds were obtained from the Korea Chemical Bank (chembank.org (accessed on 17 December 2024), Daejeon, Republic of Korea) according to the material transfer agreement, and are presented in Table 1. Among the seven compounds, F09 is a compound that has been selected in the discovery of other protein phosphatase inhibitors [45]. H08, G07, and G11 share a similar chemical backbone structure, while D04 and H03 also share a similar backbone structure.

### 3.2. Determination of IC_50_ Values

The half-maximal inhibitory concentration (IC_50_) of the seven candidate compounds was determined by a dose-dependent response assay with fixed *p*NPP concentration (50 mM) or fixed amylopectin amount (45 μg) and varying inhibitor concentrations (0–0.2 mM). The IC_50_ values were calculated directly from regression curves using GraphPad PRISM software. The IC_50_ graph was fitted to the Logistic Equation.

The IC_50_ data of the *p*NPP assay for SEX4 indicate that among the seven compounds, one significantly stood out (Table 2). Compound F05 showed the smallest IC_50_ value at 5.18 ± 0.014 μM, whereas the other compounds (F09, H08, G07, G11, D04, and H03) showed values of 15.83 ± 0.018 μM, 18.69 ± 0.026 μM, 30.91 ± 0.021 μM, 25.75 ± 0.017 μM, 21.60 ± 0.008 μM, and 39.58 ± 0.016 μM, respectively.

The IC_50_ data of the amylopectin assay for SEX4 indicate that among the seven compounds, two significantly stood out (Table 2). Compound F09 showed the smallest IC_50_ value at 7.49 ± 0.016 μM, whereas compound G11 showed the second smallest IC_50_ value at 9.72 ± 0.008 μM. The other compounds (H08, F05, G07, D04, and H03) showed values of 16.49 ± 0.051 μM, 21.10 ± 0.021 μM, 38.61 ± 0.022 μM, 57.33 ± 0.34 μM, and 13.05 ± 0.024 μM, respectively.

### 3.3. Steady-State Kinetic and Inhibition Constants

The steady-state kinetic parameters (V_max_, K_M_, and *k*_cat_ values) of the seven selected compounds were determined by varying substrate concentrations (*p*NPP 0.5–50 mM; amylopectin 5–45 μg) in the absence of an inhibitor and in the presence of an inhibitor at the IC_50_ concentration. Based on the V_max_ and K_M_ values, we determined for each inhibition model and calculated the K_i_ values of the seven selected compounds.

The V_max_ value of F09 in the *p*NPP assay was decreased, and the K_M_ value was increased in the mixed-model inhibition model; both the Vmax and K_M_ values of the other compounds were decreased in the uncompetitive inhibition model (Table 2). The K_i_ values were derived based on the inhibition models of the seven candidate compounds, as presented in Table 2. The results showed that the smallest K_i_ value was 6.87 ± 0.007 μM for compound F05, consistent with its IC_50_ value. However, the Ki values of the other compounds were around 20~45 μM. K_i_ value is the dissociation constant of the enzyme–inhibitor complex, which means the affinity of the inhibitor with the inhibitor binding site. Therefore, F05 may bind more tightly to SEX4.

The V_max_ and K_M_ values of F09, H08, F05, and G07 in the amylopectin assay were decreased in the uncompetitive inhibition model; the K_M_ values of other compounds were decreased in the competitive inhibition model (Table 2). The K_i_ values were derived based on the inhibition models of the seven candidate compounds, as presented in Table 2. The results showed that the smallest K_i_ value was 7.81 ± 0.005 μM for compound G11, whereas compound F09 showed the second smallest K_i_ value at 16.42 ± 0.012 μM. However, the Ki values of the other compounds were around 20~47 μM. Therefore, G11 may bind more tightly to SEX4.

### 3.4. Inhibitor Selectivity Between SEX4 and LSF2

IC_50_ values were determined using the *p*NPP assay and the amylopectin assay to identify a selective or specific inhibitor for SEX4 through comparison with LSF2. In the results of the *p*NPP assay for LSF2 (Table 3), no compounds showed notably distinctive IC_50_ values. However, when comparing the IC_50_ values of LSF2 and SEX4, the IC_50_ value of F05 stood out most in the *p*NPP assay for SEX4, being approximately four times lower than that of LSF2. The IC_50_ values of the other compounds were similar between SEX4 and LSF2.

The amylopectin assay was used to determine the IC_50_ values of LSF2 because amylopectin is a more actual and biological substrate than *p*NPP. In the results for LSF2 (Table 3), the IC50 values exceeded the maximum measurable concentration of the inhibitors. A comparative analysis of selected compounds with SEX4 and LSF2 confirmed that the IC_50_ values of F05, F09, and G11 were lower for SEX4 than for LSF2. These results indicate that the inhibitory compounds (F05, F09, and G11) are selective for SEX4.

## 4. Discussion

Although knockdown and knockout have significantly advanced our understanding of protein functions, these methods have limitations in capturing the rapid kinetic deviations of phosphatases and elucidating the specific effects of phosphorylation events [46]. In contrast, small-molecule inhibitors are ideal tools for uncovering their mechanisms, substrates, and diverse functions within complex networks [46]. The present study aimed to investigate chemical inhibitors of SEX4 activity and identified F09, F05, and G11 as potential inhibitors, which proved effective in both the amylopectin and *p*NPP assays.

This study utilized the *p*NPP and amylopectin assays to determine each inhibitor’s steady-state kinetic parameters (V_max_, K_M_, and *k*_cat_) and inhibition constants (IC_50_ and K_i_). Inhibition models can be classified into four types: competitive inhibition, uncompetitive inhibition, noncompetitive inhibition, and mixed-model inhibition. V_max_ represents the maximum velocity of the enzyme, while K_M_ is the dissociation constant of the enzyme–substrate complex. An increase in the dissociation constant of the enzyme–substrate complex indicates a lower affinity between the enzyme and substrate, meaning that a rising K_M_ value corresponds to a decreasing affinity. In the competitive inhibition model, only the K_M_ value is affected. In the uncompetitive inhibition model, both V_max_ and K_M_ values are influenced. In the noncompetitive inhibition model, only the Vmax value decreases. In mixed-model inhibition, both V_max_ and K_M_ values depend on the binding types.

F09 exhibited a mixed-model inhibition pattern, while the remaining six inhibitors were confirmed to follow an uncompetitive inhibition model based on the *p*NPP analysis. Therefore, the seven selected compounds may bind to a site distinct from the active site of SEX4. Among these, F05 demonstrated the lowest IC_50_ and K_i_ values in the uncompetitive inhibition model through the *p*NPP assay compared to the other compounds. *p*NPP is the simplest form of a phosphatase substrate, and the loss of activity in the *p*NPP assay primarily results from disrupting the active site of the protein phosphatase. Thus, the inhibition by F05 likely arises from its binding to a site different from the active site of SEX4. H08, G07, G11, D04, and H03 also follow an uncompetitive inhibition model, so their inhibitory effects occur by binding to a site other than the catalytic site of the SEX4 protein.

In contrast, F09, H08, F05, and G07 showed uncompetitive inhibition in the amylopectin assay, while G11, D04, and H03 exhibited competitive inhibition. These data suggest that F09, H08, F05, and G07 may bind to a site distinct from the active site of SEX4, whereas G11, D04, and H03 likely bind to the active site. Notably, G11 had the lowest IC_50_ and K_i_ values in the competitive inhibition model through the amylopectin assay compared to the other compounds. Since amylopectin is the biological substrate for SEX4 and LSF2, the inhibitions by G11, D04, and H03 are likely due to its binding to the active site of SEX4. In contrast, the inhibitions by F09, H08, F05, and G07 may result from binding to a different site from the catalytic site of SEX4.

The inhibition models of G11, D04, and H03 differ between the results of *p*NPP and amylopectin assay measurements. This seems to be based on the fact that *p*NPP is a non-biological and small-molecular-weight substrate for dephosphorylation enzymes, whereas amylopectin is a biological and high-molecular-weight substrate. In other words, amylopectin has a much larger molecular weight than *p*NPP, so it may bind to a region other than the active site of the SEX4 protein. It is presumed that G11, D04, and H03 inhibit the binding of amylopectin to SEX4, thereby reducing the SEX4 activity for amylopectin.

The inhibition models of H08, F05, and G07 are uncompetitive for the *p*NPP and amylopectin assay measurements. This suggests that these chemicals bind to sites other than the active site of the SEX4 protein and exhibit inhibitory effects. In other words, it is assumed that H08, F05, and G07 bind to an allosteric site of SEX4 and reduce SEX4 activity toward *p*NPP and amylopectin. The inhibition of F09 is mixed-model for *p*NPP assay and uncompetitive for the amylopectin assay. Mixed-model inhibition is a combined type of competitive, uncompetitive, and noncompetitive [47]. The inhibition by F09 may be also originated from binding to an allosteric site of SEX4 both *p*NPP and amylopectin assays.

The three-dimensional structures of SEX4 (Protein Data Bank identification code (PDB ID): 3NME and 4PYH), LSF2 (PDB ID: 4KYQ and 4KYR), and Laforin (PDB ID: 4R30 and 4RKK) were presented and revealed in several articles within the past 15 years [9,48,49,50,51]. The representative structural images of SEX4, LSF2, and Laforin are shown in the Appendix A. Both SEX4 and Laforin have a Carbohydrate Binding Module (CBM) domain and a DSP domain, whereas LSF2 only has a DSP domain. Overall, there is a structural similarity between the DSP domain of SEX4, LSF2, and Laforin, and the CBM domain of SEX4 and Laforin. The complex structure of SEX4 bound to a glucan chain and phosphate [49] suggests the importance of the binding pocket of the CBM domain, which is highly conserved among SEX4 orthologs.

When the chemical structures of the seven compounds obtained were applied to the three-dimensional structure of the SEX4 protein, H08, G07, and G11 were found to have a glucan-like structure, so it is thought that their inhibitory effect is shown by their binding to the CBM binding pocket. F09 has a relatively low molecular weight, so its inhibitory effect is probably due to binding to or reacting with a specific site on the SEX4 protein. The inhibitory effect of F05 is probably predicted to bind to the hydrophobic binding pocket of the SEX4 protein, and D04 and H03 have similar structures, and, structurally, their inhibitory effect is presumed to be due to binding of the ring structure to the binding pocket of the CBM domain of SEX4. To confirm this idea, we are currently conducting virtual screening using molecular dynamics and docking studies. However, the results obtained currently do not match the previously obtained kinetic results, so it seems that a more precise approach and optimization process are necessary. In addition, in order to reasonably match the virtual screening results and the in vitro biochemical analysis results, it is essential to experimentally determine the binding structure of a protein and its inhibitor. Accordingly, it is disclosed that efforts to determine the complex structures of the SEX4 protein and the inhibitors are currently in progress.

To identify selective inhibitors for SEX4, we compared the IC_50_ values of the seven selected compounds in both SEX4 and LSF2 using the *p*NPP and amylopectin assays. The results showed that the IC_50_ value of F05 was significantly lower in the *p*NPP assay for SEX4 than that for LSF2, suggesting that F05 is more selective for SEX4. In the amylopectin assay, we could not determine the IC_50_ values of the seven compounds for LSF2 due to concentration limitations. The IC_50_ values for SEX4 were lower than those for LSF2, indicating that the seven selected compounds could be selective inhibitors of SEX4 compared to LSF2 in the amylopectin assay. Looking at the three-dimensional structures of LSF2 and SEX4, these results suggest that the inhibitors that primarily show inhibitory effects by binding to the CBM binding pocket do not show inhibitory effects because LSF2 does not have a CBM domain. Therefore, it seems reasonable to target the CBM domain for the development of selective inhibitors between SEX4 and LSF2.

Based on the results of the biochemical activity measurements for selective inhibitors, it is very important and essential to confirm the direct inhibitory effect on plants using the obtained inhibitors for practical and biological verification. We obtained preliminary results by treating potatoes with the seven inhibitors obtained as a component of fertilizer. According to these preliminary results, the starch content of potatoes was found to have increased by more than 30% compared to the control, but, similar to the results of genetic mutation experiments in Arabidopsis [35,44], it showed a side effect of potato growth being reduced to about 70%. This suggests that the optimization of the inhibitor treatment time, method, dosage, etc., is needed to increase the starch content of potatoes without inhibiting growth using the inhibitor obtained to inhibit starch decomposition. For this optimization, joint research is currently underway with botanists using the discovered inhibitor.

In order to increase starch production, efforts to find specific inhibitors or regulators for proteins related to starch metabolism are very necessary, and efforts to find specific inhibitors are especially important from the perspective of the inhibition of starch decomposition. As part of these efforts, the process of searching for and confirming inhibitors for SEX4, which plays an important role in the starch decomposition process, was conducted through this study. This study is currently in its initial stages, and it is thought that some careful research and consideration are needed to expand it further and make it a feasible study.

First, it is believed that it is necessary to confirm the effect of reducing starch by practically applying it to plants such as rice, sweet potatoes, and potatoes. In this process, there are side effects that inhibit plant growth, so it is believed that an optimization process for the inhibitor treatment method, concentration, and timing to overcome this is necessary. In addition to the current primary target plant, potatoes, research on increasing the straw starch content of rice by applying it to rice is also considered important in terms of increasing biofuel production.

Second, it is judged that optimization research on the inhibitors obtained so far should be conducted. The virtual exploration process should be focused on the three-dimensional structures of SEX4, LSF2, and Laforin obtained so far, and quantitative structure–activity relationship analysis (QSAR) studies should be conducted based on this. Based on the results of these studies, it is believed that inhibitors with maximized inhibition effects and minimal side effects should be discovered.

In the inhibitor optimization research, it is necessary to first discover a specific inhibitor that has no specificity for similar glucan dephosphorylation enzymes like LSF2 and has strong selectivity for SEX4. It would probably be reasonable to conduct research based on the differences in domain structures and three-dimensional structures between LSF2 and SEX4. In addition, SEX4 seems to have a similar structure to Laforin, so it is very important to consider this. If Laforin activity is inhibited, Lafora bodies are produced in the brain, causing Lafora disease [3,6,38,50,52,53,54,55,56,57]. As a countermeasure, it is very important to explore differentiated inhibitors that utilize the structural differences between SEX4 and Laforin.

## 5. Conclusions

Our study identified F09, H08, F05, G07, G11, D04, and H03 as potential inhibitors of SEX4 activity through *p*NPP and amylopectin assays. Among these, F05 demonstrated the strongest inhibitory effect with the lowest IC_50_ and K_i_ values in the *p*NPP assay. F09, H08, F05, and G07 acted as uncompetitive inhibitors, while G11, D04, and H03 functioned as competitive inhibitors in the amylopectin assays. These compounds exhibited a selective inhibition of SEX4 over LSF2, highlighting their potential for further refinement through QSAR studies. The research on inhibiting SEX4, a key enzyme in starch degradation, is in its early stages. Future research should concentrate on elucidating their mechanisms and leveraging structural approaches to develop specific ligands for SEX4 and LSF2. To increase starch production, we should focus on a practical application to test inhibitors on crops like rice, potatoes, and sweet potatoes, the optimization of inhibitors using QSAR studies to develop specific inhibitors with minimal side effects, the design of inhibitors that differentiate between SEX4, LSF2, and Laforin avoiding off-target effects, and an investigation on biofuel production to explore the potential for increased starch content in rice straw.

## Figures and Tables

**Figure 1 life-14-01686-f001:**
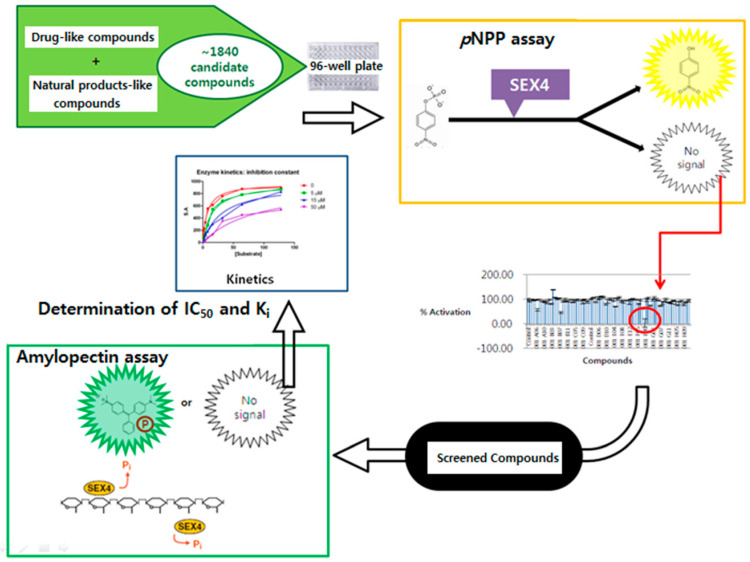
Scheme of inhibitor screening for SEX4.

**Table 1 life-14-01686-t001:** Chemical name and structure of the seven candidate compounds.

Compounds *	Chemical Name	Chemical Structure
F09	Naphthazarin	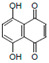
H08	Moracenin B	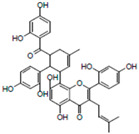
F05	Ginkgolic acid	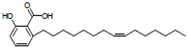
G07	Penta-O-gallolyl-β-D-glucose	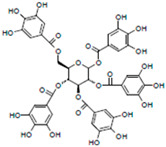
G11	Sennoside	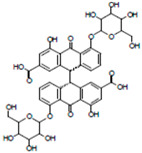
D04	Thiazolo[4,5-b]pyridin-2(3H)-one, 5-hydroxy-7-methyl-6-propyl	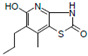
H03	Thiazolo[4,5-b]pyridin-2(3H)-one, 5-hydroxy-6,7-dimethyl	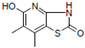

* Compound names were assigned based on their respective plate position.

**Table 2 life-14-01686-t002:** IC_50_, Ki values, and inhibition models for SEX4 in *p*NPP and amylopectin assays for selected compounds.

Compounds	*p*NPP Assay			Amylopectin Assay		
	IC_50_ (μM)	K_i_ (μM)	Inhibition Model *	IC_50_ (μM)	K_i_ (μM)	Inhibition Model *
F09	15.83 ± 0.018	39.81 ± 0.026	Mixed-model	7.49 ± 0.016	16.42 ± 0.012	Uncompetitive
H08	18.69 ± 0.026	20.84 ± 0.023	Uncompetitive	16.49 ± 0.051	31.12 ± 0.027	Uncompetitive
F05	5.18 ± 0.014	6.87 ± 0.007	Uncompetitive	21.10 ± 0.021	20.81 ± 0.021	Uncompetitive
G07	30.91 ± 0.021	37.31 ± 0.026	Uncompetitive	38.61 ± 0.022	46.40 ± 0.032	Uncompetitive
G11	25.75 ± 0.017	33.87 ± 0.019	Uncompetitive	9.72 ± 0.008	7.81 ± 0.005	Competitive
D04	21.60 ± 0.008	25.41 ± 0.012	Uncompetitive	57.33 ± 0.34	28.31 ± 0.014	Competitive
H03	39.58 ± 0.016	42.34 ± 0.037	Uncompetitive	13.05 ± 0.024	29.84 ± 0.025	Competitive

* The inhibition models were determined by steady-state kinetic measurements. The data are averages ± the standard errors of three measurements.

**Table 3 life-14-01686-t003:** Comparison of IC_50_ values between SEX4 and LSF2 through pNPP assay and amylopectin assay.

Compounds	pNPP Assay		Amylopectin Assay	
	SEX4 (μM)	LSF2 (μM) *	SEX4 (μM)	LSF2 (μM) *
F09	15.83 ± 0.018	23.97 ± 0.028	7.49 ± 0.016	>50 ± 0.21
H08	18.69 ± 0.026	22.92 ± 0.025	16.49 ± 0.051	>50 ± 0.27
F05	5.18 ± 0.014	21.26 ± 0.008	21.10 ± 0.021	>50 ± 0.31
G07	30.91 ± 0.021	13.39 ± 0.036	38.61 ± 0.022	>50 ± 0.38
G11	25.75 ± 0.017	23.03 ± 0.029	9.72 ± 0.008	>50 ± 0.52
D04	21.60 ± 0.008	44.82 ± 0.022	57.33 ± 0.34	>202 ± 0.24
H03	39.58 ± 0.016	49.58 ± 0.047	13.05 ± 0.024	>202 ± 0.35

* The IC50 values of LSF2 through the amylopectin assay were not determined with the maximum concentration of candidate chemical compounds. The data are averages ± the standard errors of three measurements.

## Data Availability

The data supporting this article are included as Appendix A.

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
