# Peer review of "Identification of Novel Inhibitors of Starch Excess 4 (SEX4)"

_life, 2024, doi:10.3390/life14121686_

Round 1
Reviewer 1 Report
Comments and Suggestions for Authors
The authors' work is highly innovative but requires revisions before publication. Here are my detailed suggestions:
1, The manuscript title, "Identification and Characterization of Novel Inhibitors of Starch Excess 4 (SEX4) in Plants", does not correspond with the presented data, as there is no in vivo data in plants. I recommend treating Arabidopsis leaves with the identified chemical compounds and analyzing starch degradation to substantiate the relevance to plant systems.
2, The Introduction section primarily discusses starch degradation in leaves. However, it is more critical to focus on slowing starch degradation in storage organs, as this holds higher agricultural significance. I suggest the authors revise the Introduction to address this aspect.
3, To more effectively improve crop yield, it would be advantageous to enhance starch degradation efficiency in leaves at night while inhibiting starch degradation in storage organs. Did the authors identify any compounds that significantly promote SEX4 enzyme activity during the screening?
4, In the context of Δ81 Arabidopsis thaliana, what does “Δ81” signify?
5, In line 121, please provide a full list of all the chemical compounds included in the library in an appendix.
6, The data in Table S1 and Table 2 should be subjected to statistical analysis to determine significance.
7, The Discussion section should include potential applications and future challenges.
8, The Conclusions section should be more concise and avoid redundant descriptions of the results.
9, Additionally, increasing starch content is not a primary breeding objective for tomatoes crop.
Author Response
Reviewer 1
The authors' work is highly innovative but requires revisions before publication. Here are my detailed suggestions:
- Thanks for your comments.
1, The manuscript title, "Identification and Characterization of Novel Inhibitors of Starch Excess 4 (SEX4) in Plants", does not correspond with the presented data, as there is no in vivo data in plants. I recommend treating Arabidopsis leaves with the identified chemical compounds and analyzing starch degradation to substantiate the relevance to plant systems.
- We agreed with this comment. We are trying to treat potatoes with identified chemicals, and having a result showing increased amount of starch, but showing stunted growth and other side effects in potatoes with the help of a botanist. Therefore, we need to find a proper treatment method and other way to solve the side effects. It seems to take a time to get a final result. We added a sentence describing our preliminary results in line 328 to 339.
- We also change the title to Identification of Novel Inhibitors of Starch Excess 4 (SEX4), according to the reviewer’s comment.
2, The Introduction section primarily discusses starch degradation in leaves. However, it is more critical to focus on slowing starch degradation in storage organs, as this holds higher agricultural significance. I suggest the authors revise the Introduction to address this aspect.
- Yes, we added the several sentences focusing on slowing starch degradation in storage organs in line 59 to 68.
3, To more effectively improve crop yield, it would be advantageous to enhance starch degradation efficiency in leaves at night while inhibiting starch degradation in storage organs. Did the authors identify any compounds that significantly promote SEX4 enzyme activity during the screening?
- We were focusing on finding an inhibitor in this study, so we didn’t look at the activating compounds so far. It will be another challenging study with a chemical library and glucan phosphatase proteins. Thanks for your suggestion.
4, In the context of Δ81 Arabidopsis thaliana, what does “Δ81” signify?
- We changed the sentence and corrected the typo to “Arabidopsis thaliana SEX4 lacking the first 89 amino acids (Δ89-SEX4)”.
5, In line 121, please provide a full list of all the chemical compounds included in the library in an appendix.
- We did not know the chemical name of all the compounds in the library according to MTA (material transfer agreement) from Korea Chemical Bank. Therefore, we have known the chemical names of seven selected compounds after screening from Korea Chemical Bank. Please understand this situation. Instead of presenting a full list of all compounds, we are presenting the screening result in supplemental figure S1.
6, The data in Table S1 and Table 2 should be subjected to statistical analysis to determine significance.
- Thanks for the suggestion. We did three independent measurements in all the assays and calculated the averages ± the standard errors to check the confidence interval and statistical significance. The assay results without an inhibitor were compared to the results of each compounds. We didn’t compare the results between them. Therefore, we were trying to secure the statistical significance as much as possible.
7, The Discussion section should include potential applications and future challenges.
- Thanks for the suggestion. We added several sentences showing potential applications and future challenges in line 340 to 369 as many as possible.
8, The Conclusions section should be more concise and avoid redundant descriptions of the results.
- Thanks for the suggestion. We changed the conclusions in a more concise version and removed the redundant descriptions in line 371 to 385.
9, Additionally, increasing starch content is not a primary breeding objective for tomatoes crop.
- We wrote sweet potatoes instead of potatoes.
Reviewer 2 Report
Comments and Suggestions for Authors
In Result 1, it is unclear how the authors selected these 7 chemical compounds as candidate inhibitors. I recommend that the authors provide the primary data from the pNPP assay, including the dosages of the compounds, as supplementary data.
The authors mentioned the potential implications of SEX4 or LSF2 inhibitors in the agricultural industry, suggesting that these could lead to increased yield and improved food security. However, the loss of SEX4 function, which mimics the inhibitory effect of SEX4 or LSF2 inhibitors, has been shown to result in stunted plant growth. This raises a conflict that needs addressing. The authors should explain how the use of SEX4 chemical inhibitors could be beneficial in agriculture despite this issue. Additionally, citing relevant references to support this claim would strengthen the argument.
Author Response
Reviewer 2
In Result 1, it is unclear how the authors selected these 7 chemical compounds as candidate inhibitors. I recommend that the authors provide the primary data from the pNPP assay, including the dosages of the compounds, as supplementary data.
- Thanks for the suggestion. We presented the dosages of the compounds in line 128 and added the sentences showing how we selected 7 chemical compounds as candidate in line 129 to 132 and in line 145 to 155. In addition, we are providing the primary data in supplemental figure S1.
The authors mentioned the potential implications of SEX4 or LSF2 inhibitors in the agricultural industry, suggesting that these could lead to increased yield and improved food security. However, the loss of SEX4 function, which mimics the inhibitory effect of SEX4 or LSF2 inhibitors, has been shown to result in stunted plant growth. This raises a conflict that needs addressing. The authors should explain how the use of SEX4 chemical inhibitors could be beneficial in agriculture despite this issue. Additionally, citing relevant references to support this claim would strengthen the argument.
- Thanks for your comments. We are also considering the side effects of treating inhibitors. Finding a proper treatment method of inhibitors is very important and necessary. We added several sentences in introduction (line 59 to 85) and discussion (line 328 to 339, line 348 to 354). We also added several references relating to down regulation of SEX4 activity in line 59 to 85.
Reviewer 3 Report
Comments and Suggestions for Authors
This study addresses a crucial topic in agricultural biotechnology by exploring chemical inhibitors of the SEX4 enzyme, which plays a central role in starch degradation in plants. Identifying inhibitors that can selectively target SEX4 could have significant applications for increasing starch accumulation in crops, thus potentially impacting food security. The study’s focus on identifying inhibitors specifically for SEX4, rather than related enzymes like LSF2, is particularly valuable. However, given that selective inhibition of SEX4 over LSF2 is critical, would further exploration of why these inhibitors favor SEX4 at the molecular level provide a more comprehensive understanding? Additionally, are there known structural variations in SEX4 that could explain its unique reactivity with certain inhibitors?
The use of both pNPP and amylopectin substrates for inhibition assays provides a well-rounded view of SEX4 activity against simple and complex substrates. While the dual-substrate approach is justified, it raises questions about assay specificity, particularly regarding the biological relevance of pNPP. The authors could discuss in more depth how pNPP inhibition translates to amylopectin behavior in planta and whether these results can reliably predict SEX4 inhibition in a natural setting. Is there a risk that pNPP-based assays overestimate inhibitor efficacy compared to amylopectin? Further, why did the study not incorporate a planta validation of these inhibitors to corroborate the in vitro findings?
The study identifies F05, F09, and G11 as key inhibitors with selective action against SEX4, with F05 demonstrating the lowest IC50 and Ki values, highlighting its potency. These findings are valuable; however, without structural insights, it remains unclear why F05, in particular, exhibits this level of selectivity. Could incorporating preliminary structural modeling, such as docking studies, enhance our understanding of how F05 and other inhibitors interact at the binding sites? Furthermore, the IC50 values indicate inhibitory potential but do not clarify if F05 could exert sustained inhibition in a cellular context.
Given the complexity of enzyme-inhibitor interactions, structural insights are essential for comprehending the inhibition mechanisms. Including molecular docking or structural biology techniques could reveal whether these inhibitors bind to allosteric sites or the active site. The paper indicates uncompetitive inhibition for certain compounds, implying potential allosteric interactions, but does not explore this in detail. Could the authors consider experimental approaches, such as crystallography or NMR, to validate the binding interactions? Additionally, what specific structural features of SEX4 might allow these inhibitors to bind selectively over similar phosphatases like LSF2?
The study categorizes the inhibitors as competitive or uncompetitive based on substrate assays. However, given that SEX4 operates within a complex biological environment, the competitive nature of inhibitors could alter in planta due to variations in substrate availability or enzyme regulation. The interpretation of inhibition type could therefore benefit from contextual biological scenarios. How might the presence of other glucan phosphatases, natural substrate variations, or changes in starch composition influence these inhibitors' efficacy? Would additional experiments exploring SEX4's function in plant chloroplast environments with natural substrates provide insights into real-world inhibition dynamics?
This study’s potential for agricultural application is significant, as selective inhibition of SEX4 may enhance starch accumulation. However, specific guidance on how these findings might be applied to different crop species would enhance the study's impact. The inhibition data should also address the feasibility of developing these compounds into practical agricultural treatments. For instance, are there crop species with similar SEX4 enzymes where the identified inhibitors could be effective? Additionally, what are the potential ecological and agronomic implications of introducing these inhibitors, especially regarding plant growth and nutrient allocation under field conditions?
This paper concludes with promising insights into the selective inhibition of SEX4, particularly through compounds like F05, F09, and G11. Future research directions could benefit from a more focused SAR approach, emphasizing structural modifications to enhance both selectivity and stability in planta. Given the current findings, it may be valuable to prioritize certain structural characteristics or binding affinities. Could the authors propose specific structural adjustments to the leading compounds based on SAR principles to enhance efficacy? Additionally, how do they envision overcoming potential limitations, such as inhibitor degradation or off-target effects, in a real-world agricultural setting?
Based on these points of critique and the accompanying questions, strengthening and clarifying the manuscript will provide a more comprehensive and impactful presentation of the study. Addressing these aspects could enhance the scientific rigor, increase the study’s applicability, and provide a clearer understanding of the inhibitors' mechanisms and potential agricultural applications.
Author Response
Reviewer 3
This study addresses a crucial topic in agricultural biotechnology by exploring chemical inhibitors of the SEX4 enzyme, which plays a central role in starch degradation in plants. Identifying inhibitors that can selectively target SEX4 could have significant applications for increasing starch accumulation in crops, thus potentially impacting food security. The study’s focus on identifying inhibitors specifically for SEX4, rather than related enzymes like LSF2, is particularly valuable. However, given that selective inhibition of SEX4 over LSF2 is critical, would further exploration of why these inhibitors favor SEX4 at the molecular level provide a more comprehensive understanding? Additionally, are there known structural variations in SEX4 that could explain its unique reactivity with certain inhibitors?
- Thanks for your comments. We added the representative structural images of SEX4, LSF2, and Laforin in supplemental figure S2. Based on the structural images of them, selective inhibition of SEX4 over LSF2 may be originated from the presence of carbohydrate binding module domain in SEX4. Therefore, we added this structural aspect in discussion (line 290 to 299) and also described a structural explanation for the inhibition of selected compounds in line 275 to 289.
The use of both pNPP and amylopectin substrates for inhibition assays provides a well-rounded view of SEX4 activity against simple and complex substrates. While the dual-substrate approach is justified, it raises questions about assay specificity, particularly regarding the biological relevance of pNPP. The authors could discuss in more depth how pNPP inhibition translates to amylopectin behavior in planta and whether these results can reliably predict SEX4 inhibition in a natural setting. Is there a risk that pNPP-based assays overestimate inhibitor efficacy compared to amylopectin? Further, why did the study not incorporate a planta validation of these inhibitors to corroborate the in vitro findings?
- Yes, we agree to your comments. We added the more detailed interpretation of inhibition in pNPP assay and amylopectin assay in line 275 to 289.
- Yes, it is very necessary to study in plants. We are trying to treat potatoes with identified chemicals, and having a result showing increased amount of starch, but showing stunted growth and other side effects in potatoes with the help of a botanist. Therefore, we need to find a proper treatment method and other way to solve the side effects. It seems to take a time to get a final result. We added a sentence describing our preliminary results in line 328 to 339.
The study identifies F05, F09, and G11 as key inhibitors with selective action against SEX4, with F05 demonstrating the lowest IC50 and Ki values, highlighting its potency. These findings are valuable; however, without structural insights, it remains unclear why F05, in particular, exhibits this level of selectivity. Could incorporating preliminary structural modeling, such as docking studies, enhance our understanding of how F05 and other inhibitors interact at the binding sites? Furthermore, the IC50 values indicate inhibitory potential but do not clarify if F05 could exert sustained inhibition in a cellular context.
- Thanks for your comments. We are doing a docking and molecular dynamics studies, but the preliminary results are not consistent with our in vitro findings. Therefore, we are supposed to get complex structures of chemicals and SEX4 protein to solve this inconsistency. Under this situation, we added the comments relating to this issue in discussion. We also changed the preference of F05, F09, and G11 into the description of the potential of 7 selected compounds because the description of the preference of F05, F09, and G11 is too much exaggerated.
Given the complexity of enzyme-inhibitor interactions, structural insights are essential for comprehending the inhibition mechanisms. Including molecular docking or structural biology techniques could reveal whether these inhibitors bind to allosteric sites or the active site. The paper indicates uncompetitive inhibition for certain compounds, implying potential allosteric interactions, but does not explore this in detail. Could the authors consider experimental approaches, such as crystallography or NMR, to validate the binding interactions? Additionally, what specific structural features of SEX4 might allow these inhibitors to bind selectively over similar phosphatases like LSF2?
- Yes, we agreed with the comments. We added the representative structural images of SEX4, LSF2, and Laforin in supplemental figure S2. We also added the structural aspect in discussion (line 290 to 299) and also described a structural explanation for the inhibition of selected compounds in line 275 to 289. We also added the comments relating to docking study and allosteric interactions in discussion.
The study categorizes the inhibitors as competitive or uncompetitive based on substrate assays. However, given that SEX4 operates within a complex biological environment, the competitive nature of inhibitors could alter in planta due to variations in substrate availability or enzyme regulation. The interpretation of inhibition type could therefore benefit from contextual biological scenarios. How might the presence of other glucan phosphatases, natural substrate variations, or changes in starch composition influence these inhibitors' efficacy? Would additional experiments exploring SEX4's function in plant chloroplast environments with natural substrates provide insights into real-world inhibition dynamics?
- Yes, we agreed to reviewer’s comments. There is a deviation between in vitro inhibition and in vivo inhibition. However, in vitro finding could present the guide line to the in vivo study. We are trying to do optimization study and in vivo study, mentioning in discussion so as to find more reliable inhibitors.
This study’s potential for agricultural application is significant, as selective inhibition of SEX4 may enhance starch accumulation. However, specific guidance on how these findings might be applied to different crop species would enhance the study's impact. The inhibition data should also address the feasibility of developing these compounds into practical agricultural treatments. For instance, are there crop species with similar SEX4 enzymes where the identified inhibitors could be effective? Additionally, what are the potential ecological and agronomic implications of introducing these inhibitors, especially regarding plant growth and nutrient allocation under field conditions?
- Thanks for the comments. We didn’t find the cases using a chemical inhibitor, but there are several cases using genetic manipulation to increase the starch content in potatoes, rice, barley, cassava. We mentioned this aspect in discussion section. We also added describing our preliminary results of treating selected inhibitors on potatoes in line 328 to 339
This paper concludes with promising insights into the selective inhibition of SEX4, particularly through compounds like F05, F09, and G11. Future research directions could benefit from a more focused SAR approach, emphasizing structural modifications to enhance both selectivity and stability in planta. Given the current findings, it may be valuable to prioritize certain structural characteristics or binding affinities. Could the authors propose specific structural adjustments to the leading compounds based on SAR principles to enhance efficacy? Additionally, how do they envision overcoming potential limitations, such as inhibitor degradation or off-target effects, in a real-world agricultural setting?
- Thanks for your comments. We added the comments relating to SAR study, side effects, and off-target effects in discussion section, and also mentioned our preliminary study of QSAR.
Based on these points of critique and the accompanying questions, strengthening and clarifying the manuscript will provide a more comprehensive and impactful presentation of the study. Addressing these aspects could enhance the scientific rigor, increase the study’s applicability, and provide a clearer understanding of the inhibitors' mechanisms and potential agricultural applications.
- Thanks for the final comment. We rewrote manuscript and added the related sentences to strengthen and clarify based on the points as much as possible. We hope you will acknowledge the significance of our findings, despite their many limitations.